# ThickV-Stain: Unprocessed Thick Tissues Virtual Staining for Rapid Intraoperative Histology

**Lulin Shi**[*1]                                                    LSHIAO@CONNECT.UST.HK
**Xingzhong Hou**[*2]                                              HOUXINGZHONG@ICT.AC.CN
**Ivy H. M. Wong**[1]                                             HMWONGAL@CONNECT.UST.HK
**Simon C. K. Chan**[1]                                            CKCHANBQ@CONNECT.UST.HK
**Zhenghui Chen**[1]                                               ZCHENEF@CONNECT.UST.HK
**Claudia T. K. Lo**[1]                                                 KETKLOAE@UST.HK
**Terence T. W. Wong**[†1]                                                TTWWONG@UST.HK

[1] *Translational and Advanced Bioimaging Laboratory, Department of Chemical and Biological Engineering, The Hong Kong University of Science and Technology, Hong Kong, China*
[2] *State Key Laboratory of Computer Architecture, Institute of Computing Technology, Chinese Academy of Sciences, Beijing, China*

**Editors:** Accepted for publication at MIDL 2024

## Abstract

Virtual staining has shown great promise in realizing a rapid and low-cost clinical alternative for pathological examinations, eliminating the need for chemical reagents and laborious staining procedures. However, most of the previous studies mainly focus on thin slice samples, which still require tissue sectioning and are unsuitable for intraoperative use. In this paper, we propose a multi-scale model to virtually stain label-free and slide-free biological tissues, allowing hematoxylin- and eosin- (H&E) staining generation in less than a minute for an image with 100 million pixels. We name this ThickV-Stain model, specifically developed to virtually stain intricated and unprocessed thick tissues. We harness the ability of a multi-scale network to encourage the model to capture multiple-level micromorphological characteristics from low-resolution images. Experimental results highlight the advantages of our method for virtual staining on unprocessed thick samples. We also show the effectiveness of ThickV-Stain on thin sections, showing generalizability to other clinical workflows. The proposed method enables us to obtain virtually stained images from unstained samples within minutes and can be seamlessly integrated with downstream pathological analysis tasks, providing an efficient alternative scheme for intraoperative assessment as well as general pathological examination.

**Keywords:** Virtual staining, deep learning, label-free imaging, unprocessed biological tissues

## 1. Introduction

Histological examination is regarded as the gold standard for cancer diagnoses, which aims to determine whether there are any lesions or analyze the tissue microenvironment (Gurcan et al., 2009). However, routine histological staining is chemical reagent-dependent and

---

[*] Contributed equally
[†] Corresponding author

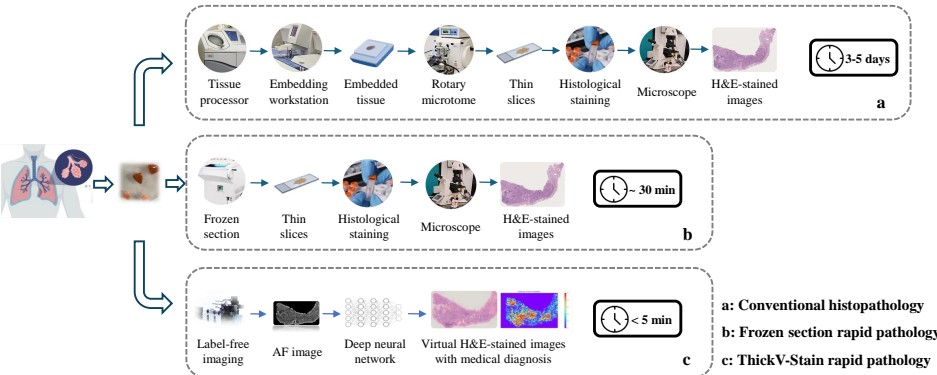

Figure 1: Histological examination workflow comparison.

usually takes several days for sample preparation, heavily limiting their applications on rapid histopathology, especially intraoperative tissue examination (Maloney et al., 2018). Although the frozen section has been considered as an alternative for rapid pathological diagnosis to guide tumor resection in surgery, frozen samples are unstable and easily introduce some artifacts during rapid freezing, making diagnosis difficult or even misdiagnosed (Taxy, 2009). With the development of section-free imaging techniques, unprocessed thick tissue samples can be visualized in grayscale images without sectioning procedures (Fereidouni et al., 2017; Glaser et al., 2017). The use of label-free or stain-free microscopes is not a routine diagnostic workflow for pathologists, and it is difficult for pathologists to interpret the biological changes with the grayscale images acquired through this method.

The emergence of virtual staining has provided a novel solution for achieving rapid histology. It refers to the digital transformation or generation of histological stains through computer algorithms (Bai et al., 2023). There are some studies on the image transformation from H&E staining to special or immunohistochemistry (IHC) staining (Liu et al., 2021; de Haan et al., 2021; Lahiani et al., 2019; Vasiljević et al., 2021; Lahiani et al., 2020; Xu et al., 2020) and image generation from unstained thin sections (Rivenson et al., 2019b,a; Zhang et al., 2020; Li et al., 2020, 2021; Meng et al., 2021; Shi et al., 2023). Although the existing virtual staining studies have achieved impressive performance, they mainly rely on well-prepared thin slices, and the acquisition and preparation of these thin slices are still cumbersome and time-consuming (as shown in Figure 1a), which is unsuitable for intraoperative tissue diagnosis. Therefore, this paper does not focus on image translation from the existing staining or unstained thin sections but focuses on virtual H&E staining from sectioning-free and label-free tissues.

In this paper, we investigate directly using autofluorescence (AF) microscopy (Zhang et al., 2022), to obtain AF images of unprocessed thick samples, which does not necessitate tissue embedding, sectioning, and dewaxing (Figure 1c). Our objective is to convert the acquired grayscale images into pathologist-interpretable images. Additionally, we evaluated that the virtually stained images can further be used for automatic tumor diagnosis. The whole procedure of the proposed pipeline can be completed within several minutes with minimal tissue preparation. In comparison to the frozen section (Figure 1b), our method is faster and non-destructive to tissues. In brief, the key contributions of this paper are as

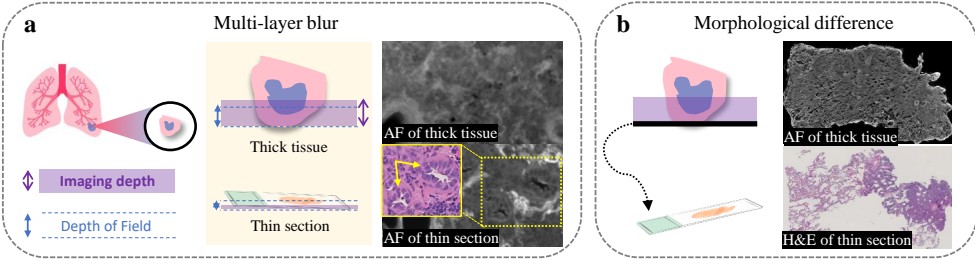

Figure 2: Comparison between thick samples and thin sections. (a) The multi-layer information due to the deeper imaging depth inevitably degrade the image quality. (b) Morphological differences between AF of thick tissue and H&E-stained image.

follows: 1). We propose a ThickV-Stain model [1] for complicated unprocessed bio-samples. 2). The first proposal to use a multi-scale generator for virtual staining from low-quality input images. 3). SOTA performance on both thick tissue and thin section data.

## 2. Method

### 2.1. Research Problem

The research problem is the automatic translation from grayscale AF images of thick samples to histologically stained images, which can be easily integrated with other downstream tasks (e.g., tumor detection, and report generation), enabling rapid intraoperative diagnosis.

The first challenge here is the low image quality of AF input. In slide-free microscopy, AF images from thick specimens suffer lower molecular contrast and are blurrier. This is due to the lack of optical sectioning capability in the slide-free imaging system and multi-layer information would be captured at the same time. From Figure 2a, we can see that the image quality and resolution of the thick samples are relatively poor when compared with that of thin tissue slices. Therefore, it is much more challenging to recognize and transform those indistinguishable components from images of unprocessed thick specimens. As shown in Figure 2a, cell nuclei arranged in adenocarcinoma acinar structure can be observed in the AF of the thin section while no obvious related features are shown in the AF of thick tissues. Hence, it poses a huge challenge to learn the correct mapping between two domains.

Another barrier is that it is not possible to obtain aligned chemically stained images with unstained images for model training as histological staining will only be carried out on a thin slide. In this situation, after acquiring scanned images of unprocessed and unstained thick samples, only the scanned surface would be sectioned and stained for reference. Since AF images of thick samples contain multiple-layer information, there will be a considerable morphological difference between the AF images and the chemically stained versions (Figure 2b). Unsupervised learning is required when there is no paired ground truth. Current unsupervised virtual staining methods mainly exploit cycleGAN as the basic architecture (Zhu et al., 2017). (Li et al., 2021) shows that the cycleGAN can hardly distinguish the background and tissue context towards the virtual staining of complicated human samples.

---

1. Code and model are available at https://github.com/TABLAB-HKUST/ThickV-Stain.

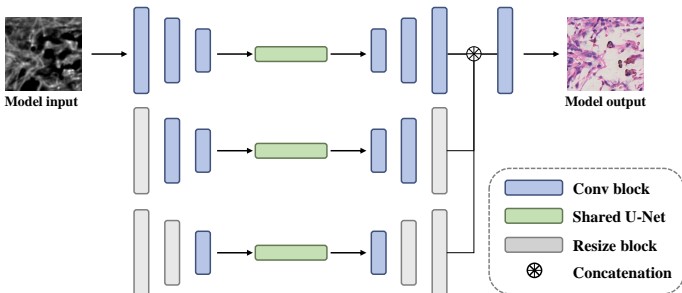

Figure 3: Overview of the proposed ThickV-Stain Generator. The multi-scale U-Nets share the same parameters.

This is because the cycle-consistency loss cannot guarantee semantic rationality. Even though the semantic change on the model output image is incorrect, the reconstructed one can be recovered perfectly due to the loss. Some work also reported that cycle consistency loss plays an indispensable role in key structure preservation of input images (Pumarola et al., 2018), but weak constraints between the translated output and ground truth (Liu et al., 2021). To resolve this ambiguity, some researchers have tried to use saliency mask (Li et al., 2021) and region labels (Shi et al., 2022) as semantic guidance for virtual staining.

In our case, it is quite challenging to extract the consistent saliency mask or region label due to the limited image contrast. We address the above challenges with the multi-scale generator. We claim to use a multi-scale model to explore sufficient information for a better understanding of the complex data, which does not require additional supervision or manual annotations. We aim to use different scales focusing on different histological patterns according to the receptive field size and capturing multi-level representations.

### 2.2. ThickV-Stain: Thick Biological Tissues Virtual Staining

To have special attention to learning multi-structural features, we propose a novel multi-scale generator as shown in Figure 3. It consists of three U-Net modules with the same convolution kernel size ($4 \times 4$) but with different receptive fields due to different input image resolutions. The top branch in Figure 3 captures more detailed histological features, such as cell nuclei. This is because the convolution kernel of this module will perceive small objects in the input resolution. Conversely, the bottom one can perceive a large field as the original image is scaled down, such as global information of edges and background.

The U-Net architecture used in this paper consists of an encoder and a decoder network. The encoder involves 5 convolution layers and the decoder consists of 5 fractionally-strided convolution layers that upsample the feature maps to the original spatial dimensions. The feature maps from the encoder with the corresponding ones from the decoder are connected by skip connections to preserve the spatial information from the input. Following the U-Net, the outputs from the two additional scales are resized to the original size of the input image and then concatenated to get the final output. The form of the output can be expressed as:

$$G(x) = conv([U_1(x), R(U_2(x)), R(U_3(x))]) \tag{1}$$

where $[\cdot]$ refers to the concatenation operator, $U_i$ represents the $i$-th branch. $R(\cdot)$ denotes the bicubic interpolation function to change the size of images. We use 2 resize blocks here to achieve the increase of two additional scales (H/2, W/2) and (H/4, W/4).

To avoid mode collapse caused by the imbalance design between the generator and discriminator, we also use a multi-scale discriminator (Wang et al., 2018b). The input image is processed at three scales, with each scale being processed by a separate network. Those three discriminator networks have an identical structure but operate at different image scales. The output of each network is then combined to produce a final score.

Besides GAN loss (Goodfellow et al., 2020), we further use identity loss (Taigman et al., 2017) for better structure preserving. The overall loss functions are shown below:

$$\mathcal{L}_{adv}^{G} = -\mathbb{E}_x[logD(G(x))] \tag{2}$$

$$\mathcal{L}_{adv}^{D} = \mathbb{E}_x[logD(G(x))] + \mathbb{E}_y[log(1 - D(y))] \tag{3}$$

$$\mathcal{L}_{idt} = \mathbb{E}_y \|y - G(y)\|_1 \tag{4}$$

$$\mathcal{L} = \mathcal{L}_{adv} + \lambda\mathcal{L}_{idt} \tag{5}$$

## 3. Experiments and results

### 3.1. Dataset

The data used in this paper were collected from 14 patients and involved various lung adenocarcinoma subtypes. The H&E-stained slices used for training were cut from the imaged thick sample surface. Due to the megapixels of whole slide images (WSIs, $\sim$20,000 $\times$ $\sim$20,000), we randomly sampled 10,000 small image patches per epoch with the size of $256 \times 256$ from seven patients for training. During testing, each WSI of the remaining seven patients' data was split into $256 \times 256$ images with 16-pixel overlap to avoid artifacts, and the test cohort included 31,317 image patches (thick sample data used in section 3.3) and 22,283 image patches (thin section data used in section 3.4).

### 3.2. Implementation Details

The overall network was implemented in PyTorch on a single NVIDIA GeForce RTX 3090 GPU. We used the architecture of U-Net in each scale forming our generator and multi-scale discriminator based on the pix2pixHD. We trained our model with the Adam optimizer (with $\beta_1 = 0.5$, $\beta_2 = 0.999$). The initial learning rate was set to $2 \times 10^{-4}$ for the generator, and $1 \times 10^{-4}$ for the discriminator with a linear decay scheduled after 15,000 iterations. The different learning rate was designed to avoid mode collapse due to the fast convergence of the discriminator. The batch size was set to 16. The $\lambda$ in 5 is set to 5.

### 3.3. Experiment on Thick Tissue Data

In this paper, we chose the commonly used unsupervised I2I architectures, including cycle-GAN, CUT, and a seminal virtual staining model (UTOM), as the baseline models. The visual comparison is shown in Figure 4. Compared to the AF image, the H&E staining ($1^{st}$ column) can only visualize some surface structure of the sample, providing a basic reference rather than an exact ground truth.

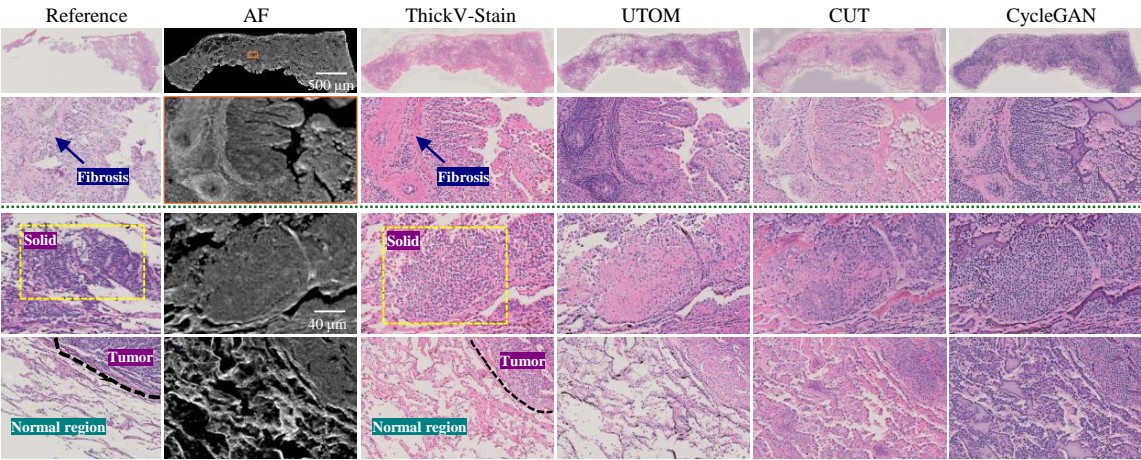

Figure 4: Comparison results on thick tissues. The $1^{st}$ column is traditional H&E staining of a thin slice sectioned from the imaged surface of thick tissue ($2^{nd}$ column).

Table 1: Quantitative evaluation with different methods.

|  |  | CycleGAN | CUT | UTOM | ThickV-Stain |
|---|---|---|---|---|---|
| **Thick** | FID↓ | 97.24 ± 5.88 | 97.26 ± 24.37 | 62.00 ± 6.88 | **36.71 ± 3.27** |
|  | KID↓ | 0.06 ± 0.01 | 0.06 ± 0.02 | 0.03 ± 0.01 | **0.01 ± 0.01** |
|  | T-MSE↓ | 1.00 ± 0.05 | **0.92 ± 0.04** | 1.06 ± 0.19 | 0.95 ± 0.02 |
| **Thin** | FID↓ | 45.21 ± 4.37 | 61.12 ± 1.39 | 95.13 ± 31.00 | **19.28 ± 1.57** |
|  | KID↓ | 0.02 ± 0.00 | 0.05 ± 0.01 | 0.05 ± 0.02 | **0.01 ± 0.00** |
|  | SSIM↑ | 0.54 ± 0.07 | 0.29 ± 0.00 | 0.46 ± 0.07 | **0.61 ± 0.02** |
|  | PSNR↑ | 14.33 ± 0.07 | 13.53 ± 0.04 | 15.74 ± 1.70 | **17.82 ± 0.07** |
|  | MSE↓ ($\times 10^2$) | 24.17 ± 0.35 | 29.03 ± 0.31 | 19.05 ± 8.11 | **11.05 ± 0.13** |

Table 2: Professional assessment average score on test slides from two pathologists. The result is rated from 0–5. Higher scores represent better quality.

|  | Visual quality | | Pathological features | | |
|---|---|---|---|---|---|
|  | **Fidelity** | **Accuracy** | **Nuclei** | **Extracellular fibrosis** | **Overall** |
| CycleGAN | 2.07 | 1.54 | 2.39 | 1.36 | 1.57 |
| CUT | 2.79 | 2.79 | 2.36 | 2.11 | 2.54 |
| UTOM | 4.11 | 2.86 | 2.29 | 2.43 | 3.00 |
| **ThickV-stain** | **4.61** | **3.43** | **2.79** | **3.18** | **3.82** |

In the $2^{nd}$ row (marked with blue arrow), there are obvious pathological features of fibrosis in our results. From the $3^{rd}$ row, the major tumor component consisting of solid sheets can be observed in our results (marked with yellow dashed lines). The same structure can also be observed in H&E-stained reference image. Even though UTOM also draws a correct outline, it misses cell nuclear information within the outline. Both CUT and cycleGAN mistranslate the background areas, which is consistent with (Li et al., 2021). To evaluate the image quality quantitatively, we calculate the Fréchet Inception Distance

(FID) (Heusel et al., 2017) and Kernel Inception Distance (KID) (Bińkowski et al., 2018). The results are shown in Table 1, where ThickV-Stain achieves the best FID score (36.71). UTOM and cycleGAN are better than CUT, showing the same trend with visual analysis.

To explore whether our virtual staining meets clinical needs, we invited two pathologists to score our virtually stained images. The evaluation metrics are from two aspects: the pathological properties and visual quality. We followed the setting from (Rivenson et al., 2019b) to evaluate nuclei details, extracellular fibrosis, and overall staining quality. Regarding the visual quality, fidelity refers to whether any artifacts in the stained image can affect the image quality and accuracy refers to whether the pathological diagnosis (e.g., the location of the tumor region) from virtually stained images is consistent with the real staining. The evaluation results in Table 2 show that our model significantly outperforms other baseline models. The results of cycleGAN and CUT provide limited useful information to the pathologist, which is consistent with our analysis of visual results.

Moreover, we apply a lung adenocarcinoma detection model for automatic tumor region detection with our virtually stained results. (Wang et al., 2018a) use human lung adenocarcinoma cancer data, which is consistent with our case. Here, we apply the trained model on both generated images and real H&E images to see if our virtual stained results can show consistent diagnostic conclusions with real staining. In addition, the MSE between the two tumor probability maps, noted T-MSE, can be used to evaluate the potential of virtually stained results in real clinical practice. The comparative results can be found in Table 1, and a visual example can be found in Appendix Figure 7.

### 3.4. Experiment on Thin Slice Data

We also validate the extensibility of the proposed method on other clinical workflows. Moreover, the ground truth is accessible for the virtual staining of thin sections, which can provide an accurate evaluation of the model performance. It can be observed that ThickV-Stain (Figure 5c) outperforms other baseline models on unstained thin slices compared with ground truth (Figure 5b). As for quantitative measures, we chose the structural similarity index measure (SSIM), peak signal-to-noise ratio (PSNR), and MSE, as metrics because ground truth can be obtained for thin-section virtual staining. The numerical results are shown in Table 1. Regarding PSNR and MSE, ThickV-Stain and UTOM are relatively close, and CUT are inferior to the other models. As for the FID and KID scores computed on the test set, ThickV-Stain significantly outperforms other models.

### 3.5. Ablation Study

In this paper, we claim that the virtual staining performance on complicated dataset, especially unprocessed thick tissue, is limited if the model cannot represent complex features adequately. To explore the importance of the multi-scale model for virtual staining of complicated thick samples, we carried out an ablation study on multi-scale generator.

From Figure 6, we can see that for virtual staining of thin section data, even a single-scale generator can also perform well. It achieves correct translation where the same cancerous patterns and fibrosis can be observed, with only slight degradation in image quality compared with the multi-scale generator. And there is no significant difference in the FID results (20.29 and 21.74). We analyze this because the AF of thin sections are of such high

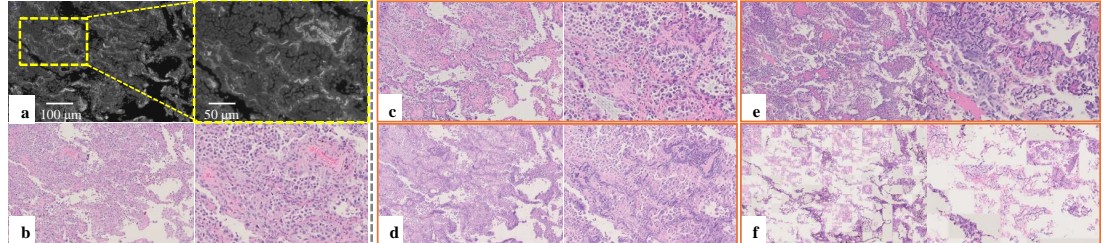

Figure 5: Virtual staining results on thin sections. (a) Input AF. (b) Real H&E staining. (c) ThickV-Stain. (d) UTOM. (e) CUT. (f) CycleGAN.

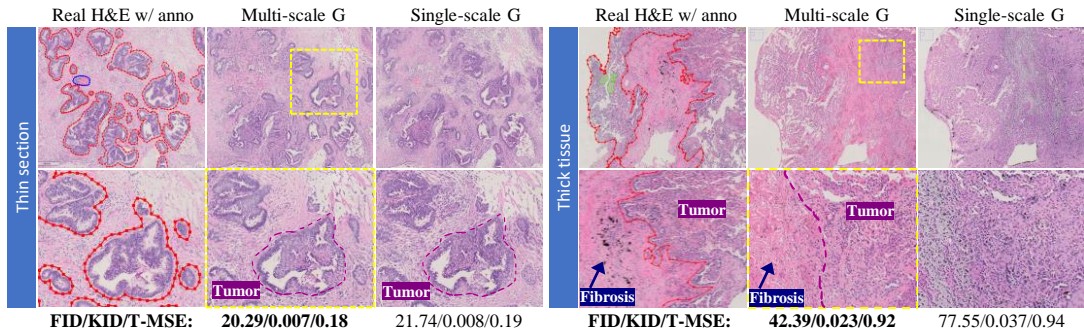

Figure 6: Ablation study. The tumor regions are marked manually with purple lines.

quality that even a single scale model is sufficient to correctly transform the image features. In this case, multi-scale does not have obvious advantages when dealing with simple data.

Secondly, for thick tissues, the single-scale generator cannot recover the fibrosis and quantitative results are also far from satisfactory compared with the multi-scale generator, which shows the importance of the multi-scale model for handling complicated data.

Additionally, the single-scale generator on thin sections even achieves better FID than the multi-scale generator on thick tissues (21.74 vs 42.39). That means even increasing the complexity of the model cannot completely offset the impact of data complexity. Virtual staining on thick tissue is still difficult to achieve comparable results on thin slice data.

## 4. Conclusion

In this paper, we present ThickV-Stain for virtual histological staining on unprocessed thick tissues. Specifically, we design a multi-scale generator to fuse large receptive field image features to achieve virtual staining on complex data. We also use a well-trained lung adenocarcinoma detection model to demonstrate the potential of our generated staining to be integrated with other downstream pathological image analyses tasks. With our proposed ThickV-Stain, the staining task of thick samples can be implemented, allowing pathologists to identify pathological features on unprocessed samples directly, thus providing an alternative to rapid pathology.

## Acknowledgments

This work was supported by the Research Grants Council of the Hong Kong Special Administrative Region (16208620 & 26203619) and an internal grant at HKUST (OKT21EG12). The authors would like to thank Dr. Ronald C. K. Chan for the valuable discussion, and Dr. Michael K. Y. Hsin for patient recruitment. The lung cancer samples collected in this paper are collected from the Queen Mary Hospital. All human experiments were carried out in conformity with a clinical research ethics review approved by the Institutional Review Board of the University of Hong Kong/ Hospital Authority Hong Kong West Cluster (HKU/HA HKW) (reference number: UW 20-335), and informed consent was obtained from all lung cancer tissue donors.

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

## Appendix A. Example of downstream task performed on virtually stained results

## Appendix B. Score table from pathologists

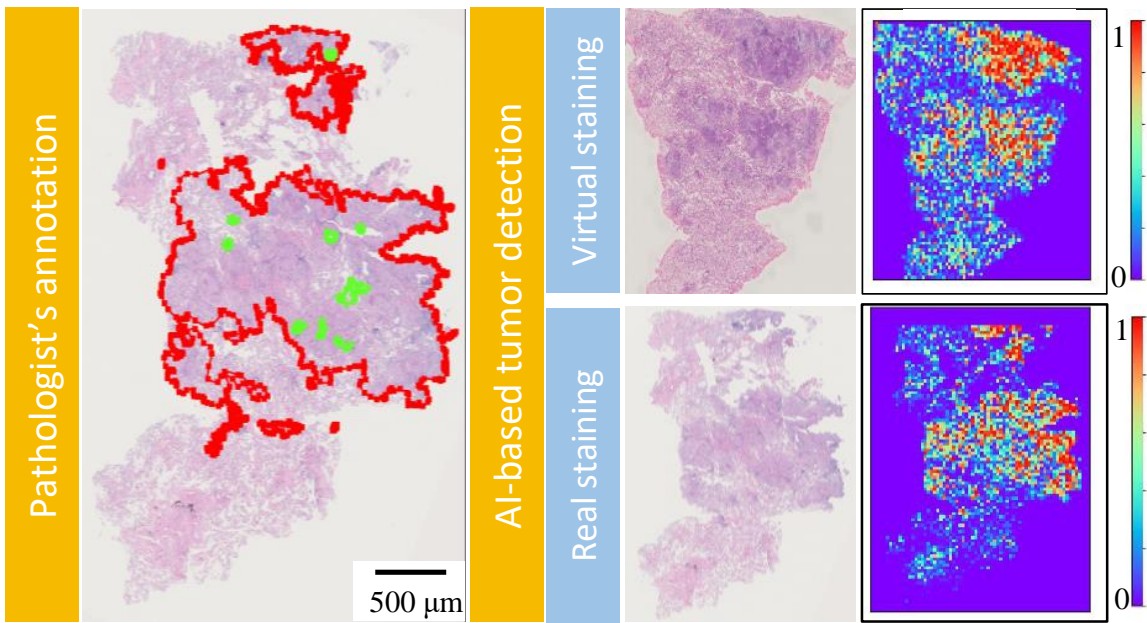

Figure 7: Comparison between pathologist's annotation and artificial intelligence (AI)-based tumor detection. The left column is tumor region annotation from pathologists for comparison, where the red lines highlight the tumor region and green areas indicate the normal region. The right part is AI-based tumor prediction results on both virtual staining and actual H&E staining, in which areas in red and blue mean a higher and lower probability of tumor, respectively.

Table 3: Raw data of score table from pathologists. a) UTOM, b) CUT, c) ThickV-Stain, d) CycleGAN

| Tissue | Pathologist 1 | | | | | Pathologist 1 | | | | | Average | | | | |
| No. | FD | AC | ND | EF | OS | FD | AC | ND | EF | OS | FD | AC | ND | EF | OS |
|---|---|---|---|---|---|---|---|---|---|---|---|---|---|---|---|
| 1(a) | 5 | 3 | 2 | 1 | 2 | 3.5 | 2.5 | 3 | 3.5 | 3 | 4.25 | 2.75 | 2.5 | 2.25 | 2.5 |
| 1(b) | 4 | 3 | 2 | 1 | 2 | 1.5 | 3.5 | 3 | 3 | 3.5 | 2.75 | 3.25 | 2.5 | 2 | 2.75 |
| 1(c) | 4 | 4 | 3 | 3 | 4 | 5 | 4 | 3.5 | 4 | 4.5 | **4.5** | **4** | **3.25** | **3.5** | **4.25** |
| 1(d) | 1 | 2 | 3 | 1 | 1 | 3.5 | 2 | 1.5 | 1.5 | 2 | 2.25 | 2 | 2.25 | 1.25 | 1.5 |
| 2(a) | 5 | 3 | 1 | 1 | 2 | 3 | 2 | 1.5 | 2.5 | 3.5 | 4 | 2.5 | 1.25 | 1.75 | 2.75 |
| 2(b) | 4 | 3 | 2 | 2 | 2 | 1.5 | 4 | 3.5 | 3.5 | 3 | 2.75 | **3.5** | 2.75 | **2.75** | 2.5 |
| 2(c) | 5 | 3 | 3 | 2 | 3 | 4 | 3.5 | 2.5 | 3 | 4 | **4.5** | 3.25 | 2.75 | 2.5 | **3.5** |
| 2(d) | 1 | 1 | 4 | 1 | 1 | 3 | 3.5 | 3.5 | 2 | 2.5 | 2 | 2.25 | **3.75** | 1.5 | 1.75 |
| 3(a) | 5 | 3 | 2 | 1 | 3 | 2 | 2 | 2 | 4 | 3.5 | 3.5 | 2.5 | 2 | 2.5 | 3.25 |
| 3(b) | 4 | 3 | 1 | 2 | 2 | 1.5 | 2 | 2 | 3.5 | 3 | 2.75 | 2.5 | 1.5 | 2.75 | 2.5 |
| 3(c) | 5 | 2 | 2 | 3 | 3 | 3.5 | 3 | 3 | 3.5 | 4 | **4.25** | 2.5 | **2.5** | 3.25 | 3.5 |
| 3(d) | 1 | 1 | 2 | 1 | 1 | 2 | 1 | 1.5 | 1.5 | 2 | 1.5 | 1 | 1.75 | 1.25 | 1.5 |
| 4(a) | 5 | 3 | 3 | 1 | 4 | 4 | 3.5 | 3 | 4.5 | 4 | 4.5 | 3.25 | 3 | 2.75 | 4 |
| 4(b) | 4 | 2 | 2 | 2 | 3 | 1.5 | 2 | 3 | 3.5 | 2.5 | 2.75 | 2 | 2.5 | 2.75 | 2.75 |
| 4(c) | 5 | 3 | 3 | 3 | 4 | 5 | 4 | 3.5 | 4.5 | 5 | **5** | **3.5** | **3.25** | **3.75** | **4.5** |
| 4(d) | 1 | 1 | 1 | 1 | 1 | 3 | 1.5 | 2 | 2 | 2 | 2 | 1.25 | 1.5 | 1.5 | 1.5 |
| 5(a) | 5 | 3 | 2 | 1 | 2 | 3 | 2 | 2 | 3 | 3 | 4 | 2.5 | 2 | 2 | 2.5 |
| 5(b) | 5 | 3 | 2 | 1 | 2 | 2 | 2.5 | 1 | 1.5 | 2 | 3.5 | 2.75 | 1.5 | 1.25 | 2 |
| 5(c) | 5 | 4 | 2 | 2 | 3 | 4 | 3.5 | 3 | 3 | 4 | **4.5** | **3.75** | **2.5** | **2.5** | **3.5** |
| 5(d) | 2 | 1 | 2 | 1 | 1 | 3 | 1 | 1.5 | 1 | 1.5 | 2.5 | 1 | 1.75 | 1 | 1.25 |
| 6(a) | 5 | 3 | 3 | 2 | 2 | 4 | 3.5 | 3 | 3.5 | 3 | 4.5 | 3.25 | 3 | 2.75 | 2.5 |
| 6(b) | 4 | 4 | 3 | 1 | 2 | 1.5 | 3.5 | 4 | 2.5 | 3.5 | 2.75 | **3.75** | **3.5** | 1.75 | 2.75 |
| 6(c) | 5 | 3 | 2 | 3 | 3 | 4 | 3.5 | 3 | 3.5 | 4 | 4.5 | 3.25 | 2.5 | **3.25** | **3.5** |
| 6(d) | 2 | 1 | 3 | 1 | 1 | 2.5 | 3 | 3 | 2.5 | 3 | 2.25 | 2 | 3 | 1.75 | 2 |
| 7(a) | 4 | 4 | 3 | 3 | 4 | 4 | 2.5 | 1.5 | 3 | 3 | 4 | 3.25 | 2.25 | 3 | 3.5 |
| 7(b) | 3 | 2 | 3 | 1 | 3 | 1.5 | 1.5 | 1.5 | 2 | 2 | 2.25 | 1.75 | 2.25 | 1.5 | 2.5 |
| 7(c) | 5 | 4 | 3 | 3 | 4 | 5 | 3.5 | 2.5 | 4 | 4 | **5** | **3.75** | 2.75 | **3.5** | **4** |
| 7(d) | 1 | 1 | 4 | 1 | 1 | 3 | 1.5 | 1.5 | 1.5 | 2 | 2 | 1.25 | 2.75 | 1.25 | 1.5 |

