# OpenReview forum: "ThickV-Stain: Unprocessed Thick Tissues Virtual Staining for Rapid Intraoperative Histology"
_MIDL.io/2024/Conference — MIDL 2024 Oral_

### Official Review · Reviewer_QhRu · 2024-02-15

**Confidence:** 5
**Preliminary Rating:** 5
**Recommendation:** Oral
**Final Rating:** 5

**Summary:**

This study introduces ThickV-Stain, an innovative method for performing virtual histological staining on unprocessed, thick tissue specimens. The approach involves the creation of a multi-scale generator that integrates image features across large receptive fields, enabling effective virtual staining on complex datasets. Furthermore, the inclusion of a sophisticated lung adenocarcinoma detection model illustrates the compatibility of their virtual staining technique with further pathological image analysis tasks. ThickV-Stain represents an advancement in the staining of thick samples, permitting pathologists to directly observe pathological features in unprocessed specimens and offering a rapid alternative for pathology evaluations.

**Strengths:**

+ The development of the ThickV-Stain model, specifically designed to tackle the challenge of applying virtual staining techniques to complex, unprocessed biological samples. This model represents a leap forward in the processing and analysis of such specimens.
+ The innovative introduction of a multi-scale generator marks the first-ever proposal in the realm of virtual histological staining, aimed at significantly improving the quality of virtual staining on low-resolution or poor-quality input images. This approach enables a more accurate and detailed visualization of biological structures.
+ Demonstrating SOTA performance across a wide range of samples, including both thick tissue specimens and thin sections, thereby broadening the applicability and effectiveness of virtual staining technologies in the field of pathology. These contributions collectively represent a substantial advancement in the field, offering new tools and methodologies for researchers and medical professionals.
+ The paper stands out for its exceptional quality, characterized by well-crafted figures and a narrative that is straightforward and engaging. It excels in presenting a clear and accessible storyline, making complex concepts easily understandable.

**Weaknesses:**

+ The study does not compare the proposed method against a comprehensive array of baseline methods. This extensive comparison would improve the robustness and rigor of the analyses.
Otherwise, I don't see any other major weakness.

**Detailed Comments:**

This is a great paper and would raise many interesting discussions in the conference.

**Justification Of Final Rating:**

I would keep my rating. The paper excels in innovation, methodology, presentation, and potential impact, making it a valuable contribution to its field and warranting a strong accept recommendation. Thanks

**Justification Of The Preliminary Rating:**

This is the highest quality paper in my review pool. This paper introduces the groundbreaking ThickV-Stain model and a pioneering multi-scale generator for virtual staining, addressing challenges with complex, unprocessed samples and low-quality images. Demonstrating state-of-the-art performance on diverse specimens, it significantly advances pathology with its clear, engaging narrative and high-quality presentation, offering valuable tools and insights for professionals.

**Questions To Address In The Rebuttal:**

N/A

**Special Issue:**

Yes

---

> ### Author Response · Authors · 2024-03-10
>
> Dear Review QhRu:
>
> Thank you for your thoughtful review and for recognizing the importance of our work.

---

### Official Review · Reviewer_94hM · 2024-02-26

**Confidence:** 3
**Preliminary Rating:** 4
**Recommendation:** Poster
**Final Rating:** 4

**Summary:**

This article addresses the issue of virtual H&E staining. The authors propose a GAN architecture dedicated to generating H&E staining images from an AF image of a thick tissue section. This task is challenging due to the blur caused by the superposition of information from the multi-layered sample. Moreover, there is no ground truth of such H&E staining as it is only performed on a thin tissue section. One of the goal of such approach is to perform rapid virtual staining in an intra-operative setting.

The novelty of the approach lies in a multi-scale generator based on three UNets. The first UNet is a classic one, while the other two use downsampled versions of the input sample to increase their receptive fields and focus on larger-scale structures.

Experiments were conducted on a private dataset of 14 patients with lung adenocarcinoma. A comparison with three baseline methods was performed on both thick and thin sections. A blind qualitative analysis was conducted by two pathologists. Additionally, the usefulness of these virtual stainings was assessed by analyzing a downstream task, namely automatic tumor region detection.
The authors claim that their approach yields superior results and enables pathologists to rapidly identify pathological features from unprocessed tissue samples.

**Strengths:**

- The article is clearly written, and it effectively establishes the context and rationale for the proposed approach.
- The authors conducted extensive quantitative and qualitative experiments, providing a thorough evaluation of their approach.
- The code to reproduce the experiments is available and seems well documented.

**Weaknesses:**

- The description of the proposed method is rather brief in the article, lacking several key details, particularly regarding the architecture of the generator. The resizing factor of the blocks and the specifics of the backbone UNet architecture are not explained. Additionally, the "shared Unet" highlighted in green in Figure 3 requires clarification. The rationale behind incorporating the identity loss should also be explained. Similarly, a concise description of the discriminator architecture is required.

- The quantitative results lack standard deviations, which could lead to questioning the significance of the findings. Including standard deviations would improve the robustness and interpretability of the results.

- Regarding the qualitative analysis, there appears to be a significant level of inter-expert variability (see Table 3), which may diminish the significance of the experiment. It would be beneficial for the authors to quantify this variability and provide commentary on its implications within the article.

**Detailed Comments:**

- In section 3.3, it appears there might be a reference error. A reference to Table 3 appears but this Table seems to be located in the appendix (if so, it should be referenced as such), and a reference to Table 2 might be more appropriate.
- Regarding Fig 7, it's unclear why pathologist annotations are not conducted on the same slide used for AI-based tumor prediction (the image in the bottom right column).
- In Fig 5, it could be interesting to rearrange the images to facilitate visual comparison. For example by placing the initial image and its results in the first row, followed by the zoomed-in version and its results in the second row.
- Fig 6 lacks clarity regarding the nature of the annotations (red, and violet contours,  fibrosis and tumor tags). Specifying whether they are manual annotations by experts or generated automatically would provide valuable context.
- The compared methods (CycleGan, CUT and UTOM) are not properly cited both in the text (section 3.3) and in the Tables. This should be corrected.

**Justification Of Final Rating:**

The authors have appropriately addressed the majority of my comments and subsequently revised their article. The only remaining minor point would be to include a sentence explaining the rationale behind the choice of identity loss, as articulated in their response.
I keep my rating and believe that the article is suitable for publication in MIDL.

**Justification Of The Preliminary Rating:**

This is a well written article tackling an interesting problem and proposing a reasonable approach. The methodological novelty of the approach is limited, however the results appear reasonable and supported by extensive experiments.
However, there is a lack of detail in describing the proposed method, particularly concerning the architecture of the generator.

**Questions To Address In The Rebuttal:**

- Add more details on the generator and discriminator architectures
- Add standard deviation on the quantitative results
- Address comments of the "detailed comments" section.

**Special Issue:**

No

---

> ### Author Response · Authors · 2024-03-10
>
> Dear Review 94hM:
>
> Thank you for your thorough review and valuable feedback on our work. We address each of them below.
> 1. Regarding the details of the model design:
>
> **Generator**: The resize block is a bicubic interpolation function to change the size of images. We use 2 resize blocks here to achieve the increase of two additional scales (H/2, W/2) and (H/4, W/4). The original image and two resized images are input to the corresponding convolution block for channel adaptation and then to the same U-Net backbone. The U-Net architecture consists of an encoder and a decoder network. The encoder network is 5 convolutional layers that downsample the input image by decreasing its spatial dimensions while increasing the number of channels. The decoder network consists of 5 fractionally-strided convolution layers that upsample the feature maps to the original spatial dimensions while decreasing the number of channels. The contracting and expanding paths are connected by skip connections, which concatenate the feature maps from the encoder network with the corresponding feature maps from the decoder network. These skip connections help preserve the spatial information from the input image and improve the image generation accuracy. Following the U-Net, the outputs from the two additional scales are resized to the original size of the input image. These three outputs are then concatenated to get a feature map with dimensions (3C, H, W). Finally, a convolutional layer is used to map the feature map from (3C, H, W) to (C, H, W) to produce the final result.
>
> **Discriminator**: we use a multi-scale discriminator which is proposed in pix2pixHD. In a multi-scale discriminator, the input image is processed at multiple scales, with each scale being processed by a separate discriminator network. Those 3 discriminator networks have an identical network structure but operate at different image scales. We downsample the real and generated images by a factor of 2 and 4 to create an image pyramid of 3 scales. The discriminators are then trained to differentiate real and synthesized images at 3 different scales, respectively. The output of each discriminator network is then combined to produce a final discriminator score.
>
> We have revised the model design in the main text and highlighted all changes .
>
> 2. Regarding the **standard deviation**:
> We added some experiments and the standard deviation results shown as below:
> |    | CycleGAN | CUT | UTOM | ThickV-Stain (ours) |
> | --------| :-----------:|:--------:| :-----------: |:-----------:|
> | FID   |97.24$\pm$5.88| 97.26$\pm$24.37| 62.00$\pm$6.88|**36.71$\pm$3.27**|
> | KID   |0.057$\pm$0.011|0.063$\pm$0.016|0.034$\pm$0.005|**0.015$\pm$0.005**|
>
> 3. The rationale of **identity loss**
>
> For unsupervised image-to-image translation, the generally used loss function is cycle consistency loss which has been confirmed to play an indispensable role in key structure preservation of input images. However, for virtual staining, some research and our previous work (see below) show that cycle consistency loss cannot guarantee the correct translation between tissue content and background. Therefore, we omit this loss for model efficiency. However, without this loss, the model will lose some original structure, and then we add the identity loss, which also can help the model preserve the input structure and will not increase the model parameters.
>
> Shuting Liu, et al. Unpaired stain transfer using pathology-consistent constrained generative adversarial networks. IEEE Transactions on Medical Imaging, 40(8):1977–1989, 2021.
>
> Xinyang Li, et al. Unsupervised content-preserving transformation for optical microscopy. Light: Science & Applications, 10(1):1–11, 2021
>
> Lulin Shi, et al. One-side virtual histological staining model for complex human samples. In 2022 IEEE-EMBS International Conference on Biomedical and Health Informatics (BHI), pages 1–4. IEEE, 2022.
>
> 4. Regarding **Fig 7**:
> This is a quite good point.  Here the pathologist annotation is directly requested from our laboratory partners. Due to the different parameters set when exporting H&E WSI, there are some color and rotation differences between the image in the bottom right and the left one. We have unified those two images in the revised pdf.
>
> 5. Regarding **Fig 6**:
> The annotations in Fig. 6 are manually labeled for interpretation of the figure. Thanks for your advice and we have emphasized this in the main text.
>
> We once again thank you for the helpful comments and suggestions. We hope we have addressed all your concerns. Please don't hesitate to let us know if we can clarify anything else!

---

### Official Review · Reviewer_yRfX · 2024-02-28

**Confidence:** 4
**Preliminary Rating:** 4
**Recommendation:** Poster

**Summary:**

This paper proposed a multi-scale generator for virtual histological staining on unprocessed thick tissues. They also evaluated detection results of a pretrained model using samples processed with proposed staining method to show method can be integrated with other downstream pathological image analyses tasks.

**Strengths:**

This paper clearly defines the problem and discusses the challenges involved, providing a solid foundation for the research. Also, the comprehensive comparison with state-of-the-art methods adds significant value, helping readers understand the effectiveness of the proposed approach. The paper additionally provides details on applying the technique to thin slices and complex datasets, showing its potential.

**Weaknesses:**

The paper should improve model explanation and elaborate on chosen evaluation metrics as well as data details. Authors are advised to provide a comprehensive comparison discussion on results for better understanding of reader and reason why proposed methods performs better than others.

**Detailed Comments:**

minor comments:

- add a,b,c to Figure 1
- what are the tissue thickness? does it have effect on the performance of model?
- Please provide additional explanation on the chosen evaluation metrics and the rationale behind their selection. Elaborate on what each metric assesses and how it facilitates comparison of model performance in staining.

**Justification Of The Preliminary Rating:**

paper clearly explains a problem and provides details on existing works and provides an approach. Then by comparing the methods with SOTA justifies the applicability of technique and its better performance.

**Questions To Address In The Rebuttal:**

minor comments

**Special Issue:**

No

---

> ### Author Response · Authors · 2024-03-10
>
> Dear Review yRfX:
>
> Thanks for your very positive review, and constructive suggestions, especially regarding the detailed model explanation and chosen evaluation metrics, we will improve it in the next version.
>
> Below is our response to your comments:
>
> 1.	Regarding **tissue thickness**:
>
> We can image the sample directly no matter how thick it is because we only image the sample surface. For microscopy, only the information within the penetration depth can be captured by the microscope and the depth of light penetration is limited (no more than 100μm). Therefore, no matter how thick the sample is, light cannot penetrate deeper, deeper information is difficult to obtain.
>
> If we only focus on the surface information of the sample, then the thickness of the sample does not matter. So it will not affect the model performance. At the same time, we do not need to process the sample (tissue embedding, tissue sectioning, etc.) and can directly image the sample, enabling rapid imaging.
>
> 2.	Regarding the **chosen evaluation metrics**:
>
> FID and KID are generally used to evaluate the image quality for generative models which does **not require paired ground truth**. This is exactly the case for this paper where there are morphological differences between the AF images (model input) and the stained versions (ground truth).  Therefore, we chose FID and KID and calculated the similarity between the two datasets of images.
>
> But for virtual staining on a thin section (~5 μm), we can prepare the ground truth because the chemical stain procedures can be performed on the same thin section that was imaged. With ground truth as reference, we further chose SSIM, PSNR, and MSE. Those three metrics are commonly used to assess the quality or fidelity of generated images when **the ground truth is provided**.
>
> Moreover, the above metrics are used to evaluate the quality of the generated images or whether the generated image is similar to the real one. To see whether the generated images meet the real clinical needs, we invited pathologists to grade images based on their expertise.
>
>
> We hope this response has addressed your concerns effectively. We are grateful for the chance to discuss our work and thank you again for your valuable input.

---

### Meta-Review · Area_Chair_Uuor · 2024-03-30

**Recommendation:** Accept (Oral)
**Confidence:** 5

**Metareview:**

The reviewers unanimously agree to accept the submission.

---

### Decision · Program_Chairs · 2024-04-05

Accept (Oral)